# Resveratrol Regulates the Expression of Genes Involved in CoQ Synthesis in Liver in Mice Fed with High Fat Diet

**DOI:** 10.3390/antiox9050431

**Published:** 2020-05-15

**Authors:** Catherine Meza-Torres, Juan Diego Hernández-Camacho, Ana Belén Cortés-Rodríguez, Luis Fang, Tung Bui Thanh, Elisabet Rodríguez-Bies, Plácido Navas, Guillermo López-Lluch

**Affiliations:** 1Centro Andaluz de Biología del Desarrollo, Universidad Pablo de Olavide-CSIC-JA, and CIBERER, Instituto de Salud Carlos III, 41013 Sevilla, Spain; mezacathemeza@gmail.com (C.M.-T.); jdhercam@upo.es (J.D.H.-C.); abcorrod@upo.es (A.B.C.-R.); tungasia82@yahoo.es (T.B.T.); ecrodbie1@upo.es (E.R.-B.); pnavas@upo.es (P.N.); 2Immunology and Molecular Biology Group, Universidad del Norte, Barranquilla 081007, Colombia; lfang@uninorte.edu.co; 3School of Medicine and Pharmacy, Vietnam National University, Hanoi 100000, Vietnam; 4Departamento de Deporte e Informática, Universidad Pablo de Olavide, 41013 Sevilla, Spain

**Keywords:** resveratrol, coenzyme Q, liver, antioxidant, mitochondria, high-fat diet

## Abstract

Resveratrol (RSV) is a bioactive natural molecule that induces antioxidant activity and increases protection against oxidative damage. RSV could be used to mitigate damages associated to metabolic diseases and aging. Particularly, RSV regulates different aspects of mitochondrial metabolism. However, no information is available about the effects of RSV on Coenzyme Q (CoQ), a central component in the mitochondrial electron transport chain. Here, we report for the first time that RSV modulates *COQ* genes and parameters associated to metabolic syndrome in mice. Mice fed with high fat diet (HFD) presented a higher weight gain, triglycerides (TGs) and cholesterol levels while RSV reverted TGs to control level but not weight or cholesterol. HFD induced a decrease of *COQs* gene mRNA level, whereas RSV reversed this decrease in most of the *COQs* genes. However, RSV did not show effect on CoQ_9_, CoQ_10_ and total CoQ levels, neither in CoQ-dependent antioxidant enzymes. HFD influenced mitochondrial dynamics and mitophagy markers. RSV modulated the levels of PINK1 and PARKIN and their ratio, indicating modulation of mitophagy. In summary, we report that RSV influences some of the metabolic adaptations of HFD affecting mitochondrial physiology while also regulates *COQs* gene expression levels in a process that can be associated with mitochondrial dynamics and turnover.

## 1. Introduction

Coenzyme Q (CoQ) is a redox lipidic component of cell membranes. It is mainly located within mitochondria but significant amounts are found in sub-cellular organelles such as lysosomes, peroxisomes, the endoplasmic reticulum and Golgi apparatus [1]. In mitochondria, CoQ is an essential component of the electron transport chain, but in all cells membranes, including mitochondria, CoQ is the main endogenous lipophilic antioxidant [2,3]. CoQ not only protects cell membrane lipids against oxidation but also maintains a redox cycle with α-tocopherol in membranes [4] and ascorbic acid [5] inside and outside cells to maintain their respective antioxidant activity. To perform this antioxidant activity, membrane linked redox enzymes such as Cytochrome b_5_ reductase (CYTB_5_R) and NAD(P)H quinone dehydrogenase 1 (NQO1) [6] transfer electrons from cytosolic NAD(P)H to CoQ to maintain high levels of the reduced form, ubiquinol, the active form [7]. Recently, it has been demonstrated that a mitochondrial protein, ferroptosis suppressor protein 1 (FSP1, formerly known as apoptosis-inducing factor mitochondrial 2, AIF2), can move from mitochondria to plasma membrane to maintain the antioxidant capacity of ubiquinol in this membrane [8,9]. Recently, a new oxidoreductase enzyme linked to the outer side of plasma membrane of hepatocytes has been discovered to reduce CoQ located in blood plasma lipoproteins [10].

CoQ is synthesized in mitochondria from where it moves to the rest of cell membranes [1]. Synthesis of CoQ is a complex process, governed by at least 13 genes that codify for COQ proteins. Proteins codified by these genes synthesize CoQ in a multistage procedure located at the inner mitochondrial membrane [11]. The dysfunction of any of these proteins reduces significantly the production of CoQ causing severe mitochondrial diseases in humans [12]. All cells have the capacity to synthesize their own CoQ although the phenotype associated with both variants and mutations in COQ proteins varies depending on the mutation and the protein affected [13]. Furthermore, many other diseases can show secondary CoQ deficiency although it is not associated with direct dysfunction in enzymes involved in CoQ biosynthesis [14].

In humans, dysregulation of CoQ_10_ levels have been associated with many diseases. Particularly heart disease has been associated with lower levels of CoQ_10_ in plasma and higher oxidation of low-density lipoproteins (LDL), increasing the risk of atherosclerosis [2]. Most of metabolic and age-related diseases have been associated with CoQ_10_ dysregulation [15,16,17,18,19]. Interestingly, CoQ_10_ can be affected during the development of non-alcoholic steatosis in rats [18] and a similar effect has been suggested in humans [17].

The stilbene polyphenol resveratrol (RSV) and many other polyphenols are bioactive compounds involved in the modulation of many physiological processes in the organism [20,21]. The antioxidant activity of RSV is based on its capacity to induce endogenous antioxidant enzymes through activation of the nuclear factor erythroid-derived 2-like 2 (NRF2) transcription factor [22,23]. Many age-related and metabolic diseases are associated with mitochondrial dysfunction and oxidative damage [24]. RSV modulates mitochondrial dynamics and turnover and antioxidants defenses [20,21,25]. This capacity can explain positive effects of RSV in aging and metabolic diseases [26,27,28,29,30].

To date, regulation of CoQ synthesis and the expression of *COQ* genes is not clear. It seems that post-transcriptional regulation plays an important role in the regulation of the mRNA levels of the components of the synthesis machinery of CoQ affecting mRNA [31] or protein lifespan [14,32]. RSV also changes mitochondrial physiology in mice under HFD conditions [20,21]. We have previously found that RSV affects CoQ levels and activity of CoQ-dependent oxidoreductases in old animals in an organ-dependent effect [28,30,33]. In the present study, we wanted to know if the RSV effects in obese animals also include the regulation of CoQ synthesis and the modulation of *COQ* genes and if this regulation is associated with the modulation of the mito/autophagy system.

## 2. Materials and Methods

### 2.1. Animals

Male C57BL/6J mice (Charles River, Écully, France) were used in this study. Twelve 4-week old animals were divided in three groups. They were housed into enriched environmental conditions in groups of 4 animals per polycarbonate cage in a colony room under a 12 h light/dark cycle, with temperature (22 ± 3 °C) and controlled humidity (50% humidity). Procedure of this study was approved by the Ethical Committee of the University Pablo de Olavide (ethical code: 30-06-14-105) and followed the international rules for animal research. After determining the weight during a period of 8 weeks, animals were assigned randomly to any of the three groups: a) standard diet, control group (*n* = 4), b) high fat diet (HFD, *n* = 4) and c) high fat diet plus resveratrol (HFD + RSV, *n* = 4). Standard diet contained 13 energy% of fat whereas HFD contained 49 energy% of fat from lard. The composition of lard used in this study showed a 24.67% of palmitic acid, a 14.48% of stearic acid and a 44.88% of oleic acid and minor compositions of others fatty acids as indicated previously [34].

The control and HFD groups were provided with water containing 0.18% ethanol used as vehicle for RSV (180 μL ethanol/100 mL H_2_O). The group treated with RSV drank water containing RSV (180 μL of 0.1 g/mL ethanolic solution of trans-RSV in ethanol/100 mL H_2_O) (Cayman Chemicals, Ann Arbor, MI, USA) from the beginning. RSV and vehicle treated water was always provided in opaque bottles to avoid light-dependent decomposition of RSV as previously indicated [29]. Bottles were changed twice a week to avoid RSV degradation. As animals drank around 4–5 mL/day and weight of HFD-fed animals was around 40 g, the calculated dose of RSV was between 720 and 900 μg/animal/day (18–22.5 mg/kg/day).

After ending of the experiment, animals were sacrificed by cervical dislocation in fasting conditions (O/N fasting) and blood was obtained by cardiac puncture immediately. Organs were immediately frozen in liquid nitrogen after dissection and maintained at −80 °C until analysis.

### 2.2. Blood Metabolites Analysis

Blood was introduced into lithium heparinized tubes and immediately centrifuged at 3000× *g* at room temperature (RT) for 10 min to recover plasma. Plasma was divided into aliquots and stored at −80 °C until analysis. Total Cholesterol was determined by enzymatic analysis (Chol Reflotron: 10745065) and Triglycerides were determined using a specific kit (QCA S.A. Ref: 992320). Total antioxidant status (TAS) in plasma was determined by using a specific kit following the instructions of the fabricant (Cayman Chemicals, 709001, Ann Arbor, MI, USA).

### 2.3. Slot Blotting for Detection of Protein Carbonyl Groups

Protein carbonylation was performed as indicated by Tung et al. [30], based on a combination of 2,4-dinitrophenylhydrazine derivatization and dot blotting. Slot blot detection was developed using an enhanced chemiluminescence detection substrate Immobilon Western Chemiluminescent HRP Substrate. Carbonylated proteins were visualized by the ChemiDoc XRS+ System and compiled with Image Lab 4.0.1 Software (Bio-Rad Laboratories, Hercules, CA, USA) for quantification.

### 2.4. Mitochondria Isolation and Western Blotting

Mitochondria were extracted from liver as indicated previously [14]. Liver was finely chopped and homogenized with a Teflon-glass potter in 10 volumes of 20 mM 2-[4-(2-hydroxyethyl)piperazin-1-yl]ethanesulfonic acid (HEPES), 225 mM sucrose, 75 mM mannitol, 1 mM ethylene glycol-bis(β-aminoethyl ether)-*N*,*N*,*N*′,*N*′-tetraacetic acid) (EGTA) pH 7.4, on ice. The homogenate was centrifuged at 1000× *g* for 10 min at 4 °C to remove nuclei and unbroken debris. This homogenate was used as whole homogenate for protein determinations. Supernatant was then centrifuged at 15,000× *g* for 10 min at 4 °C. Pellet was considered as crude liver mitochondria. Protein was determined by Bradford’s method and after dissolved in loading buffer (LB), equal amounts of protein homogenates were separated on a PAGE-SDS gel and transferred onto a nitrocellulose membrane (Millipore). Ponceau S staining was used to monitor transfer efficiency and quantification of total loading. After blocking membrane with 5% skim milk dissolved in 0.5 mM Tris–HCl (pH 7.5), 150 mM NaCl, and 0.05 % Tween-20 for 1 h at RT, membranes were incubated overnight with the commercial primary antibodies indicated in Table 1. For anti-CYTB_5_R determinations we used a rabbit polyclonal antibody gently provided by J.M. Villalba group [35]. After washing with Tris-buffered saline with 0.1% Tween-20 (TBST), blots were incubated with secondary goat anti-rabbit IgG H&L (HRP) (Abcam ab6721), goat anti-mouse IgG-HRP (Sigma A4416, Barcelona, Spain), donkey anti-goat-HRP (SantaCruz sc2020, Dallas, TX, USA) 1:3000 for 1 h at RT. After washing protein was determined by using Immobilon Crescendo Western HRP Substrate (Merck Millipore, Barcelona, Spain) and visualized with a ChemiDoc™ MP Imaging System (Bio-Rad Laboratories, Hercules, CA, USA) and proteins quantified with Image Lab™ 4.0.1 Software (Bio-Rad Laboratories, Hercules, CA, USA).

### 2.5. RNA Expression Determination

Total mRNA was obtained from liver samples after homogenization with TRIzol (Invitrogen, Life Technologies, Carlsbad, CA, USA) followed by extraction with RNeasy Mini kit (Qiagen Iberia S.L, Madrid, Spain). Possible contaminant DNA was cleaned with RNAse free DNAse I (Sigma Aldrich, Barcelona, Spain). Quantity and quality of the total RNA obtained was determined by a NanoDrop ND-1000 UV spectrophotometer (Thermo Scientific, Waltham, MA, USA). RNA was used to obtain cDNA using the iScript cDNA synthesis system (Bio-Rad USA, Hercules, CA, USA) using an iCYcler Thermal Cycler (Bio-Rad, Hercules, CA, USA) according to the manufacturer’s instructions. Specific mRNA levels were determined by using quantitative qPCR using a CFX Connect Real-Time PCR Detection System (Bio-Rad, Hercules, CA, USA) from oligonucleotides designed in this regard and obtained from Eurofins MWG Synthesis GmnH (Ebersberg, Germany) (Table 2) [36] using the iTaq Universal SYBR Green Supermix (Bio-Rad, Hercules, CA, USA). To determine the best housekeeping gene used for reference we compared the levels with four constitutive genes: b-actin, HSP90, HPRT and 18S. To select the best gene we used the RefFinder software [37,38].

### 2.6. CoQ_10_ Determinations

CoQ were extracted and determined from liver homogenate as indicated previously [39]. One hundred pmol CoQ_6_ as was used as internal control. Sodium dodecyl sulphate (1%) was added to the sample and vortexed immediately during 1 min. Ethanol:isopropanol (95:5) in a proportion 2:1 was added and mixed again during 1 min. For organic extraction, 600 μL hexane were added to the mixture and vortexed again during 1 min. After a centrifugation at 1000× *g* for 10 min 4 °C, the upper organic phase was removed and stored. Organic extraction was repeated twice, and all the upper organic phases mixed. CoQ-containing organic phase was dried with speed-vac at 35 °C. After that, dried lipid extract was dissolved in 60 μL ethanol and injected in duplicate in HPLC with a 20 μL loop. Lipid components were separated by a HPLC system Beckman 166-126 (Beckman Coulter, Brea, CA, USA) equipped with a 15-cm Kromasil C-18 column (Sigma Aldrich, Barcelona, Spain) maintained at 40 °C in a flux of 1 mL/min of mobile phase of 65:35 methanol/2-propanol y 1.42 mM lithium perchlorate. Total levels of CoQ were detected by an electrochemical detector and expressed as pmol/mg protein.

### 2.7. Statistical Analysis

Statistical analysis was performed using Sigma Plot 12.5 (Systat Sofware Inc., San Jose, CA, USA) and GraphPrism 6.01 software (GraphPad, San Diego, CA, USA). Data is indicated as the mean ± SD. Comparison between two groups was performed by using the Two-tailed unpaired Student’s *t* test applying the Shapiro-Wilk normality test. Analysis of more than two groups was performed by One-way ANOVA test accompanied by a Bonferroni post-hoc analysis applying the Kolmogorov–Smirnov normality test. Pearson’s correlation analysis was used to determine the relationship between two parameters and a two-tailed *p* value was considered with a 95% confidence interval. Statistical significance was determined with a *p* < or equal to 0.05.

## 3. Results

### 3.1. Resveratrol Did Not Reverse Weight Gain in HFD Fed Animals

In our model, animals fed with HFD increased weight during the 9 months of experiment. RSV did not reduce this increase (Figure 1A) neither reduced the weight gain at the end of the experiment (Figure 1B). The plasmatic parameters associated with metabolic syndrome status of the animals indicated that HFD group showed higher TGs and cholesterol than control group (Figure 1C,D). Interestingly, RSV was able to decrease TGs to control levels but unable to decrease total cholesterol in these animals. Regarding antioxidant capacity in plasma, HFD did not affected TAS in plasma and RSV did not produce any effect (Figure 1E). However, in the case of protein carbonylation in plasma, RSV did reduce the levels significantly in comparison with both control and HFD groups (Figure 1F).

### 3.2. Levels of mRNA of Component of the CoQ Synthome in Liver Are Affected by HFD and Reverted by Resveratrol

The expression of the components of the synthesis of CoQ (the CoQ-synthome) can be affected by aging [36]. In order to determine if the expression of these components was affected by HFD in liver of these animals we performed qPCR determination of their levels in comparison with β-actin. In general, the relative expression of all the components of CoQ synthesis was reduced by HFD (Figure 2). Only in some cases, *COQ6* and *COQ8/ADCK3*, the levels in relationship with the control were clearly the same. In some cases, with respect to *COQ2* and *COQ4*, the decrease was not statistically significant although the tendency to decrease was clear.

Addition of RSV to HFD reverted this decrease affecting most of the components, except *COQ3*, *COQ7* and *COQ10*. In the case of *DPS1* and especially in *COQ8A/ADCK3*, the effect of resveratrol produced an increase of the levels of mRNA above the levels found in control group.

All these cases indicate that the expression of the genes involved in the synthesis of CoQ in liver was downregulated by HFD and that RSV can revert this effect restoring the levels to control levels although not in all the cases.

### 3.3. Levels of COQ Proteins Are Not Affected by the Changes in CoQ-Synthesis Genes Expression

In order to determine if the reduction in the expression of *COQ* genes was affecting the levels of the proteins, COQ4 and COQ7 proteins, which showed different response to HFD and HFD+RSV were detected in crude mitochondrial extracts (Figure 3). Surprisingly, no changes in the levels of proteins were found in these extracts. COQ7 protein showed a light trend to decrease in HFD+RSV treatment but without reaching significance. In the case of COQ4, the variation of levels depicted a similar trend than mRNA levels but without showing any statistical significance. This indicates that the regulation at mRNA expression does not represent the behavior of protein levels in liver mitochondria.

### 3.4. Levels of CoQ Are Not Affected by the Changes in CoQ-Synthesis Genes Expression

Levels of CoQ in liver from the different groups did not changed by HFD or HFD + RSV (Figure 4). Variation in the levels of CoQ_9_, the main form in rodents, or CoQ_10_, only showed a tendency to increase with RSV in the case of CoQ_10_ but without reaching significance. Interestingly this trend was associated with a decrease in the ratio of CoQ_9_/CoQ_10_. This tendency could be associated with a higher need for antioxidant function since. In mice, we can speculate that CoQ_10_ is associated with antioxidant function.

### 3.5. CoQ-Dependent Antioxidant Enzymes Did Not Respond to HFD or RSV

RSV is known to induce antioxidant activities by activating NRF2 pathway [40,41]. The presence of CYTB_5_R and NQO1, to known CoQ-dependent antioxidant enzymes, was not affected neither by HFD nor by RSV supplementation to HFD-fed animals (Figure 5A,B). However, surprisingly, HFD decreased the levels of mRNA for both antioxidant enzymes, and RSV was unable to reverse it to normal levels as it did with most of the *COQ* genes.

### 3.6. Mitochondrial Dynamics and Mitophagy Markers Are Affected by RSV

In order to determine if modifications in mitochondrial dynamics can be associated with the regulation the expression of *COQ* genes, we determined the presence of markers of fusion and fission in crude mitochondrial extracts (MFN2, OPA1, DRP1) and markers of mitophagy (PINK1, PARKIN and BNIP3L/NIX) (Figure 6). Our results demonstrate that RSV treatment of HFD-fed animals show a tendency to increase the fusion marker MFN2 but without reaching significance. No increase of markers of fission, DRP1, was found. Interestingly, RSV decreases the levels of PARKIN at the same time that increases the levels of PINK1 indicating a putative regulation of mitophagy. In fact, the ratio PINK1/PARKIN was higher in HFD group and was reduced to control levels when RSV was added. The induction of mitophagy in HFD and further induced by RSV was more evident with the clear increase of levels of BNIP3L/NIX in crude mitochondria samples.

In order to determine if the changes in the markers of mitochondrial dynamics and mitophagy was accompanied by higher autophagy markers, we check the levels of BECLIN 1 and Ubiquitin-like-conjugating enzyme ATG3 (ATG 3) in whole homogenates. We found that both markers of autophagy increased in comparison with control levels and RSV further increased the levels of autophagy related 3 (ATG3) protein (Figure 7). To find out if mitochondrial mass was affected in parallel with the increase in autophagic markers, VDAC1 levels were also determined in whole homogenate. In both, HFD and HFD-RSV homogenates, levels of VDAC1 were significantly lower in comparison with controls indicating the activation of autophagy in these conditions (Figure 7).

## 4. Discussion

RSV increases longevity in animals fed with high fat diets [20]. However, it was unable to produce the same prolongevity effect in animals fed with standard diet although RSV did improved many metabolic parameters [25]. In HFD conditions, RSV reduced liver, heart, and vascular pathology in an effect associated with the reduction of the mitochondrial dysfunction [20,21,25]. In agreement with these studies, in our study, RSV treatment did not decrease body weight although the profile of plasma TGs decreased without affecting cholesterol levels [20,42]. Polyphenols have been proposed to reduce several manifestations of metabolic syndrome including dyslipidemia in an effect associated with the balance of oxidoreductase enzymes, regulation of antioxidant signaling pathways and the restoration of the mitochondrial function [43]. In our study, we introduce a new aspect of this modulation, the regulation of the expression of the genes involved in the synthesis of CoQ that are repressed in HFD conditions.

RSV has shown clear activity to influence mitochondrial function in many models showing mitochondrial dysfunction [20,21,25,33,42]. Besides the above indicated metabolic effects, RSV was able to protect mitochondria against damage in a model of Complex I deficiency (NDUFC2 deficiency), under high salt stress [44]. Further, RSV and other polyphenols also protect mitochondrial dysfunction induced by indomethacin in Caco2 cells [45]. In humans, RSV has been proposed to reduce metabolic dysfunction in many diseases in which mitochondrial dysfunction is involved such as type II diabetes, cardiovascular or hepatic diseases [42,46,47].

Although CoQ is an essential component of the electron transport chain in mitochondria and its deficiency produces dramatic effects on mitochondrial physiology [48,49], its relationship with RSV or other polyphenols is only based on the combination of supplements. The clearest relationship of RSV with CoQ was found when RSV was considered a putative precursor for CoQ synthesis in yeast [50]. However, in animals, and particularly in mammals, this fact seems to be very unlikely and RSV cannot be considered a source of the phenol head of CoQ [32,50]. Furthermore, RSV and CoQ have also shown negative interactions since RSV has been shown to interfere in the transit of CoQ-delivered electrons between mitochondrial complexes I and III [51]. To our knowledge, this is the first study that shows that RSV can influence the expression of genes involved in CoQ synthesis.

RSV has been shown to induce antioxidant response against oxidative damage through activation of NRF2 [23,40]. NRF2 is a transcription factor that not only modulates several antioxidant proteins but also mitochondrial physiology. RSV has been shown to activate NRF2 reducing oxidative stress and mitochondrial dysfunction [40]. Particularly, NRF2 regulates mitochondrial biogenesis, quality and redox homeostasis through mitochondrial antioxidant enzymes, mitochondrial transcription factors and mitophagy [52]). In fact, RSV could reverse some of the effects of HFD activation through epigenetic modulation of the NRF2 promotor [53]. If the modulation of *COQ* genes mRNA levels is modulated by NRF2 remains to be clarified since no information is available to date. However, HFD is known to repress NRF2 signal [54] and the reduction of *NOQ1* mRNA levels in our experiment confirm this effect. On the other hand, the null effect of RSV in *NQO1* mRNA levels indicate that the modulation of the mRNA of most of the *COQ* genes does not depend on NRF2 activity. We cannot affirm the same in the case of *COQ3, COQ7* or *COQ10* that show the same lack of response to RSV. 

Another interesting aspect of COQ proteins are the discrepancies between the levels of mRNA and protein levels that can be considered a hallmark of this system. Several studies have demonstrated that mRNA levels of *COQ* genes do not correspond with the levels of their respective proteins [55,56,57,58]. In *COQ9* KO mutants, *COQ7*, *COQ8A/ADCK3*, *COQ5* and *COQ6* barely showed significant modifications at the mRNA level, however, at the protein level, COQ7 and COQ5 showed a great decline in kidney and muscle whereas COQ8A/ADCK3 and COQ6 showed a significant increase. In mice models of caloric restriction, minor increases in the expression of some *COQs* genes were accompanied with small decreases in the levels of their respective proteins [58]. To this complex scenario, we have to add the differences in the pattern of expression of *COQs* genes among different tissues and organs [36,58]. In agreement with this complex relationship, a study carried out in a model of secondary CoQ deficiency in mouse concluded that changes in the mitoproteome do not reflect necessarily the changes at the transcriptional levels [59]. In the case of *COQ* genes, COQ7 protein levels depend more on mRNA stability than on mRNA expression modulation [31], and we have recently found that mitochondrial proteases can severely affect the levels of COQs proteins [14] adding more complexity to the regulation of COQ proteins. 

CoQ levels have been associated with mitochondrial dynamics associated with mitofusins 2 (MFN2) [60]. MFN2 is involved in mitochondrial fusion [61] and tethering of mitochondria to the endoplasmic reticulum [62]. We consider that increases in mitochondrial fusion induced by RSV and modulation of mitophagy can probably affect mitochondrial dynamics and increase CoQ turnover without affecting total CoQ levels and proteins. This fact could explain the increase in mRNA levels without affecting protein levels of COQ and total CoQ levels. 

Our results indicate that PINK1/PARKIN ratio increases in HFD fed animals whereas RSV decreases this ratio to control levels. In healthy mitochondria, PINK1 is not stabilized in the outer mitochondrial membrane through cleavage with PARL, a mitochondrial protease that regulates the levels of some COQs members [14]. In healthy mitochondria, PARL cuts PINK1 and avoids mitophagy but, in damaged mitochondria, reduction of PARL activity ends in the attachment of PINK1 to mitochondrial membrane and the recruitment of PARKIN that activates mitophagy [63]. Our results indicate that HFD increases PINK1 levels whereas RSV tend to decrease them at the same time that increase PARKIN levels. The higher ratio PINK1/PARKIN found in HFD liver can indicate an unbalance in the mitophagy process that is modulated by RSV treatment (Figure 6). Induction of mitophagy by RSV is also reinforced by the induction of the presence of BNIP3L/NIX, an essential factor in the induction of the autophagic machinery [64], in mitochondrial extracts (Figure 7). These evidence together with the induction of ATG3 protein [65] and BECLIN 1 [66], two known mito/autophagy-related proteins, indicate that RSV could accelerate mitophagy in liver in HFD-fed animals. We postulate that regulation of mitophagy can be associated with CoQ turnover in cells and probably COQ proteins and that *COQ* genes mRNA levels can be associated more with mitochondrial turnover regulation than with mitochondrial dynamics. 

Our study has been carried out in young animals, but it would be interesting to perform similar studies in older animals that already show modifications in antioxidant activities including CYTB_5_R and NQO1 and that respond to RSV treatment [30]. In old animals fed with a standard diet, RSV supplementation increased NQO1 activity and Glutathione Peroxidase 1 activity and protein levels although it was unable to affect CYTB_5_R [30]. Further, in mice, the influence of RSV in antioxidant activities was age and tissue dependent [28]. Thus, if RSV modulates *COQ* gene mRNA and protein levels, it is possible that its effect will be more important in old than in young animals. This same age related difference has been already found with caloric restriction or exercise in rodents affecting liver and other tissues [28,67,68,69].

## 5. Conclusions

In summary, RSV administration to mice on HFD could regulate some parameters associated to HFD such as triglycerides but no other such as weight gain or cholesterol. Mitochondrial turnover is influenced by HFD and RSV could regulate it to control levels. We show here for the first time that, HFD regulates expression levels of *COQs* genes inducing a decrease whereas RSV recover the mRNA levels of many of them to control levels. Our results indicate that mRNA levels of *COQs* genes can be associated with the rate of mitochondrial turnover whereas CoQ levels and proteins are more stable. Further studies are needed in other tissues and organs and in older animals in order to determine if the effect of RSV is age and organ-dependent. 

## Figures and Tables

**Figure 1 antioxidants-09-00431-f001:**
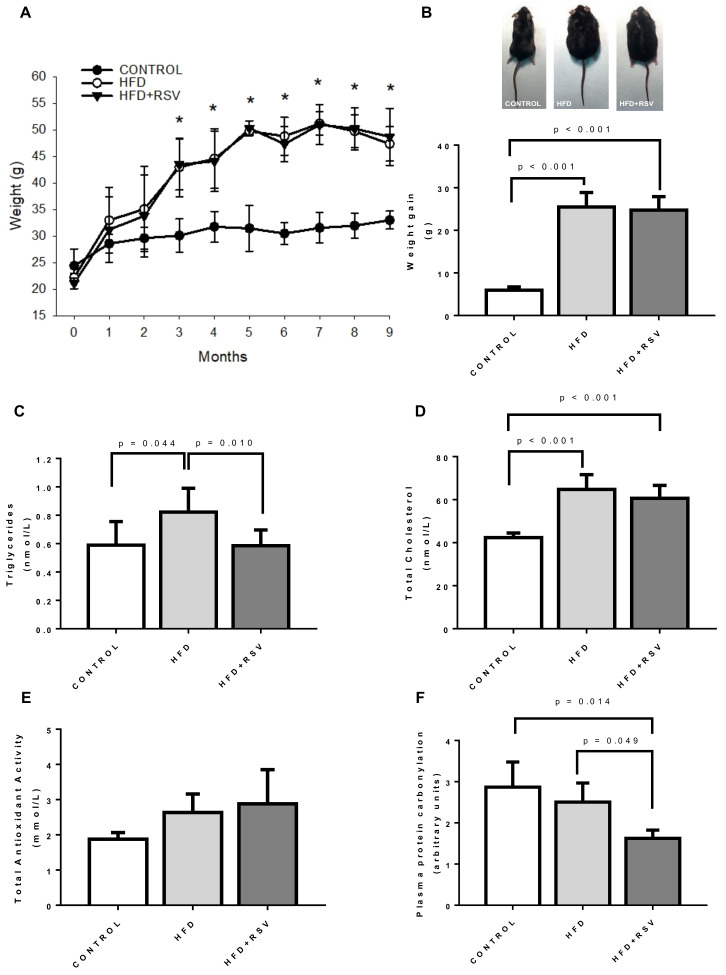
Markers of metabolic syndrome in mice. (**A**) Weight along experiment, * Statistical difference vs. control group, *p* < 0.05. (**B**) Weight gain at the end of 9 months of experiment and representative picture of animals. (**C**) Triglycerides levels in plasma (nmol/L). (**D**) Total cholesterol levels in plasma (nmol/L). (**E**) Total antioxidant status (TAS) in plasma (mmol/L). (**F**) Plasma protein carbonylation levels in liver (arbitrary units). Statistical significance is indicated in each figure. HFD: high-fat diet; RSV: resveratrol.

**Figure 2 antioxidants-09-00431-f002:**
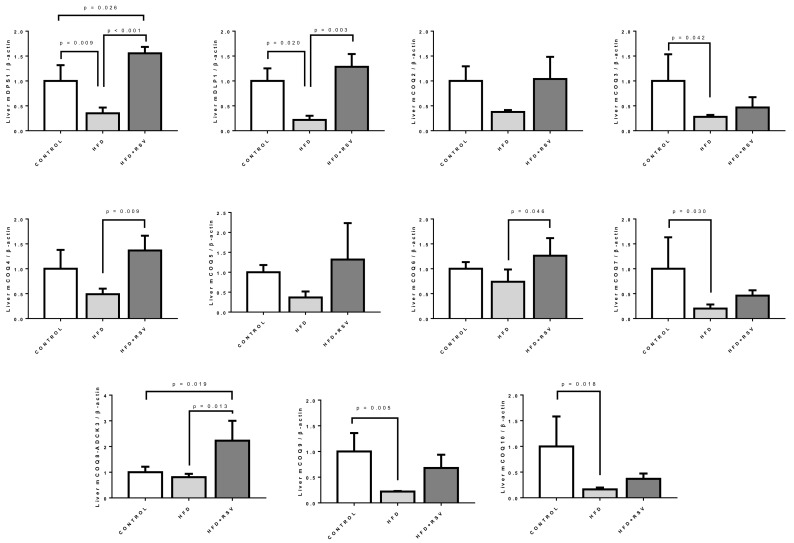
mRNA levels of COQs genes in liver. Messenger RNA levels were determined by qPCR as indicated in Section 2.5. The housekeeping gene β-actin was used as loading control. Statistical differences are included in each figure (*n* = 4) per group. mDPS1: Decaprenyl-diphosphate synthase subunit 1; mDLP1: Decaprenyl-diphosphate synthase subunit 2; mCOQ2–mCOQ10: mouse COQs proteins.

**Figure 3 antioxidants-09-00431-f003:**
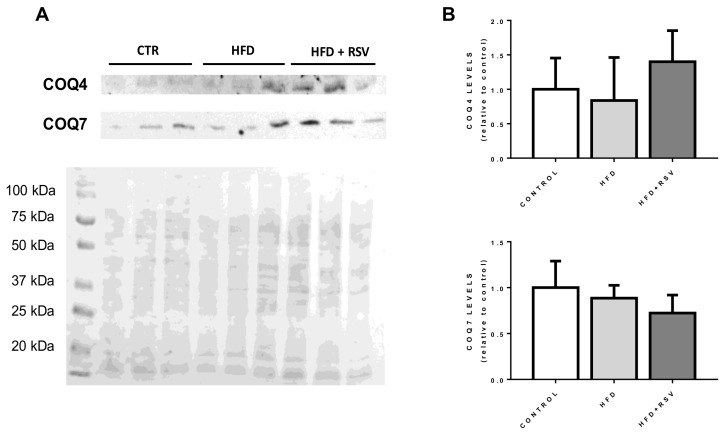
Protein levels of COQ4 and COQ7 in liver mitochondria. (**A**) Representative western blottings for COQ4 and COQ7 proteins in mitochondrial extract of liver mitochondria and Ponceau S loading marker. Lanes include the samples from different animals belonging to the three groups. (**B**) Quantification of WB relative to control levels (*n* = 4). CTR: control.

**Figure 4 antioxidants-09-00431-f004:**
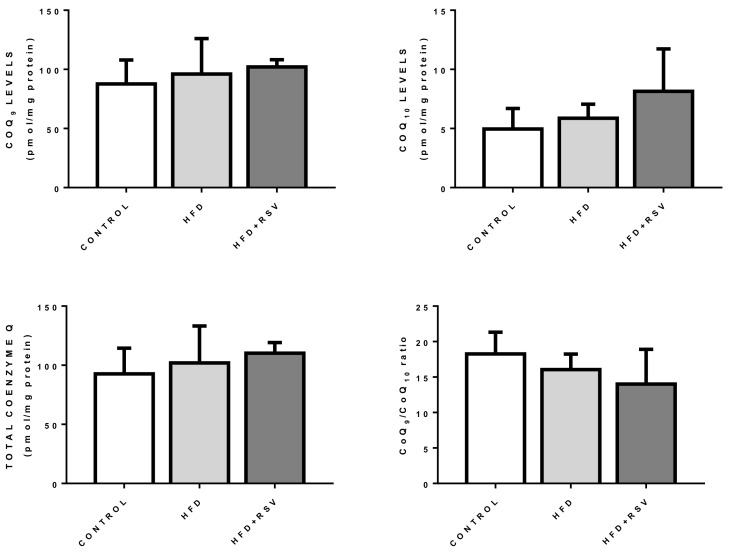
CoQ_9_ and CoQ_10_ levels in mouse liver. Figure represent the levels of CoQ_9_, CoQ_10_, total CoQ and ratio CoQ_9_/_10_ in whole homogenate of liver (*n* = 4).

**Figure 5 antioxidants-09-00431-f005:**
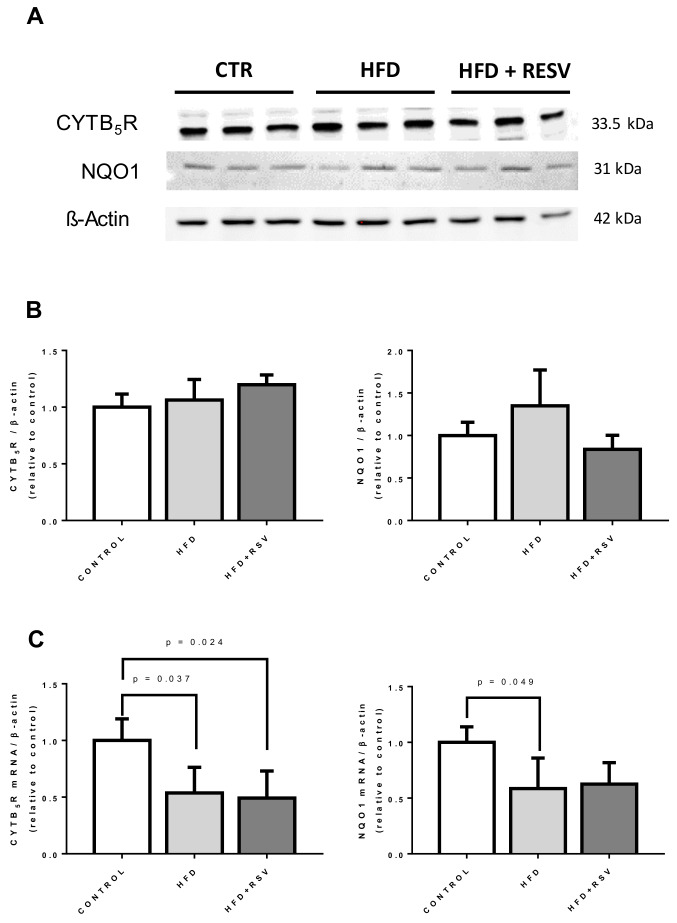
Protein and mRNA levels of CoQ-dependent oxidoreductases in mouse liver. (**A**) Representative western blotting for CYTB_5_R and NQO1 and β-actin in whole liver homogenate from three different animals per group. (**B**) Quantification of protein levels (*n* = 4). (**C**) Levels of mRNA of indicated genes. Statistical significance is indicated in each figure (*n* = 4).

**Figure 6 antioxidants-09-00431-f006:**
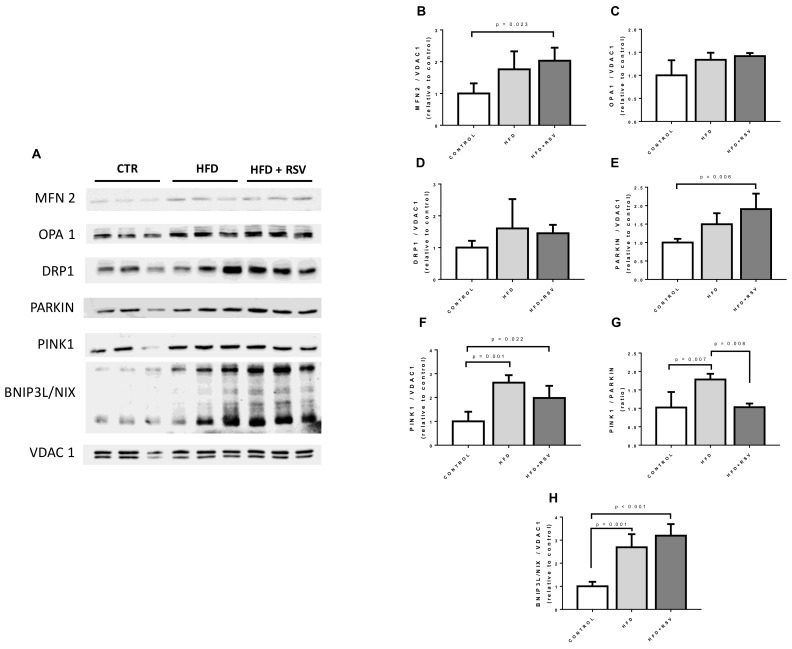
Mitochondrial turnover and dynamics markers in crude liver mitochondria. (**A**) Representative western blottings for mitochondrial markers in crude mitochondria samples from three different animals per group. (**B**–**H**) Quantification of the proteins referred to Voltage-Dependent Anion-selective Channel protein 1 (VDAC1) (relative to control). Statistical significance is indicated in each figure (*n* = 4). MFN2: mitofusin 2; OPA1: dynamin-like 120 KDa protein; DRP1: dynamin-1-like protein; BNIP3L/NIX: BCL2/adenovirus E1B 19 kDa protein-interacting protein 3-like.

**Figure 7 antioxidants-09-00431-f007:**
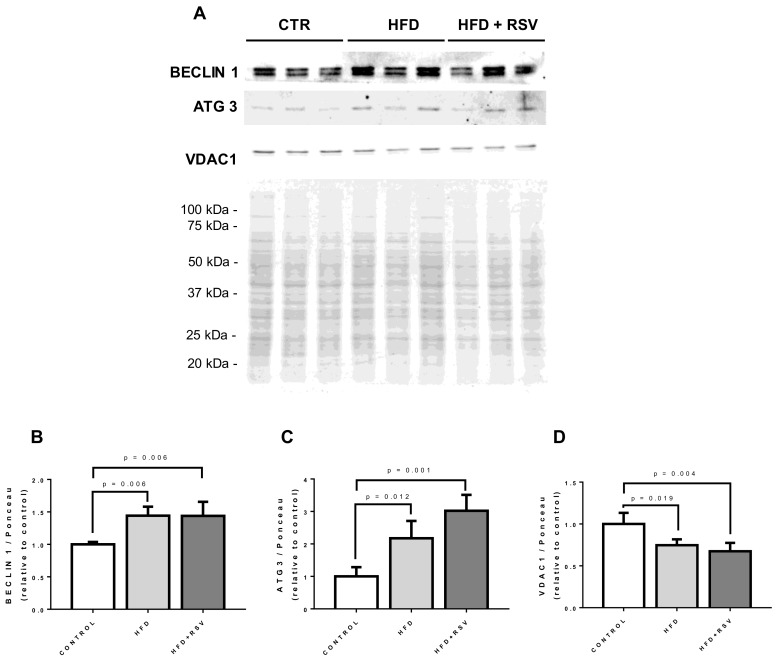
Autophagy markers in whole homogenate. (**A**) Representative western blottings for autophagy markers in whole liver homogenates from three different animals per group. (**B**–**D**) Quantification of the proteins referred to Ponceau S as loading control (relative to control). Statistical significance is indicated in each figure (*n* = 4). ATG3: ubiquitin-like-conjugating enzyme ATG3.

**Table 1 antioxidants-09-00431-t001:** Primary antibodies used in this study.

Primary Antibody	Host	Brand (Code)	Dilution
Anti-COQ4	Rabbit	Proteintech (16654-1-AP)	1:1000
Anti-COQ7	Rabbit	Proteintech (15083-1-A)	1.1000
Anti-VDAC1	Rabbit	Abcam (ab50838)	1:5000
Anti-MFN2	Mouse	Abcam (ab56889)	1:2000
Anti-OPA1	Rabbit	Abcam (ab157457)	1:1000
Anti-DRP1	Rabbit	Abcam (ab184247)	1:1000
Anti-PARKIN	Mouse	Abcam (ab77924)	1.1000
Anti-PINK1	Rabbit	Abcam (ab23707)	1:1000
Anti-BNIP3L/NIX	Rabbit	Cell Signaling (12396S)	1:1000
Anti-BECLIN 1	Rabbit	Cell Signaling (/3738S)	1:1000
Anti ATG3	Rabbit	Cell Signaling (3415)	1:1000
Anti-CYTB_5_R	Rabbit	Navarro et al. [35]	1:500
Anti-NQO1	Goat	Santacruz (sc-16464)	1:200
Anti-β-actin	Mouse	Origene (TA811000S)	1:5000

**Table 2 antioxidants-09-00431-t002:** Primers used in this study.

Gene	Forward (5′–3′)	Reverse (5′–3′)
*mDPS1*	5′-CATCAAAGGACACCAGCAATGT-3′	5′-GCACCACAATAATCGGTCTAAAGG-3′
*mDLP1*	5′-ATGCTGACCTCCAGCCTTTT-3′	5′-GTCACACCTTTGCCAGCTTT-3′
*mCOQ2*	5′-GCCCACCAGCAGGACAAGAAAGAC-3′	5′-AGCCACAGCAGCGTAGTAGG-3′
*mCOQ3*	5′-GTGAGCCACCTGGAAATGTT-3′	5′-CCCACGTATGAGTGCCTTTT-3′
*mCOQ4*	5′-GGGGAGACCACAGGATGC-3′	5′-GTCGAGGGTAGACAGCGAGAT-3′
*mCOQ5*	5′-GGATTCCTTGGGAGGTTCA-3′	5′-GGGCAGTTCTTCAGCGTCT-3′
*mCOQ6*	5′-CGACGTGGTGGTGTCAGC-3′	5′-AGTTTCTCCAGGGCTTTCTTT-3′
*mCOQ7*	5′-TGATGGAAGAGGACCCTGAGAAG-3′	5′-GCCTGTATCGTGGTGTTCAAGC-3′
*mCOQ8/ADCK3*	5′-AGCAAGCCACACAAGCAGATG-3′	5′-CCAGACCTACAGCCAGACCTC-3′
*mCOQ9*	5′-CCCGAGTTTTCCCGTCC-3′	5′-TGGGCTCCTTCAGCAATG-3′
*mCOQ10*	5′-TAAACAGAACCCTTCCACCG-3′	5′-CGAAATGCTGATAGTCCTCCA-3′
β-actin	5′-TGACCGAGCGTGGCTACAG-3′	5′-GGGCAACATAGCACAGCTTCT-3′
mHSP90	5′-GTGCCTGGAGCTCTTCTCC-3′	5′-CGTCGGTTAGTGGAATCTTCAT-3′
mHPRT	5′-CAGTCAACGGGGGACATAAA-3′	5′-AGAGGTCCTTTTCACCAGCAA-3′
m18S	5′- TGACTCAACACGGGAAACCT-3′	5′-AACCAGACAAATCGCTCCAC-3′

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
