# Peer review of "Resveratrol Regulates the Expression of Genes Involved in CoQ Synthesis in Liver in Mice Fed with High Fat Diet"

_antioxidants, 2020, doi:10.3390/antiox9050431_

Round 1
Reviewer 1 Report
The manuscript “Resveratrol regulates the expression of genes involved in CoQ synthesis in liver in mice fed with high fat diet” by Meza‐Torres et al. describes the influence of supplementation with resveratrol on some metabolic adaptations to high fat diet and regulation of CoQs genes expression in mice. The article meets the scope of Antioxidants and could be potentially interesting to the readers of this journal. However, there are some major and minor corrections that authors should implement in order to improve the quality of their work.
Major concerns:
- Figure 3, 5 and 6, what three bands in CTR, HDF and HDF+RSV represent? Are those replicates? If so, there were 12 animals involved in the study, divided in three groups, which gives 4 animals per group. Please explain, what the bands are representing.
- Figure 5A, please explain what “TRI” and “TRI+RESV” stands for. Please be consistent with abbreviations used, and explain all new abbreviations first time they appear in the manuscript.
- Figure 5, please provide number of replicates in the description.
Minor concerns:
- Please delete the period at the end of the manuscript title (page 1, line 4).
- Line 18, in the sentence, there is “again”, should be “against”, please correct.
- Abbreviation for Coenzyme Q sometimes shows up as “CoQ” and sometimes as “COQ”. Please be consistent and implement corrections.
- Line 56 and line 59, shows up the same generic statement “many diseases”. It doesn’t read well, please correct.
- Line 65 “Polyphenols and resveratrol…” this statement is misleading as it suggests that resveratrol is not a polyphenol. Please re-write.
- Lines 76-79, this sentence is too long and doesn’t read well, therefore the meaning is lost. Please re-write.
- Line 194, the abbreviation TAS is not explain. Please explain all non-obvious abbreviations fist time, when they show up in text. Moreover, this abbreviation refers to Figure 1E, “Total antioxidant activity”, shouldn’t it be “total antioxidant status”?
- Figure 2, the wording in the sentence “B-actin was used as a housekeeping gene” is unfortunate, since B-actin is housekeeping gene, and was used as a loading control in this study. Please re-word.
- Starting from line 225 the new abbreviation “HDF” shows up. Do the authors mean high fat diet (HFD)? Again, abbreviations should be consistent. Please correct or explain new abbreviation.
- Line 240, what does “se” mean? Please correct.
- Figure 6, please explain, what HDF stands for.
Author Response
Thank you for the indication about the interest of our study. We agree that this study is quite interesting for Antioxidant readers. Regarding your comments, we have introduced some modifications:
1.- Figure 3, 5 and 6, what three bands in CTR, HDF and HDF+RSV represent? Are those replicates? If so, there were 12 animals involved in the study, divided in three groups, which gives 4 animals per group. Please explain, what the bands are representing.
Response: In each western blotting determination, we ran in each line the sample from a different animal. In images, we show a representative WB in which three different samples from each group are included. Quantification was perfomed taken into consideration four animals per group, from other WBs. We have introduced this information in the text both in Material and Methods section and in figure legend.
2.- Figure 5A, please explain what “TRI” and “TRI+RESV” stands for. Please be consistent with abbreviations used, and explain all new abbreviations first time they appear in the manuscript.
Response: This is a mistake, we apologize. We included an internal older indication for the samples by mistake. TRI means triglicerydes for the HFD fed animals, however we discarded this name when we started to write the manuscript. However, in this image, the older nomenclature remained. Sorry about that.
3. Figure 5, please provide number of replicates in the description.
Response: As in other images (3, 5 and 6), the number of replicates has been introduced in the text and the clarification about the representative western blotting.
Minor concerns:
1. Please delete the period at the end of the manuscript title (page 1, line 4).
Response: thanks, dot has been eliminated.
2. Line 18, in the sentence, there is “again”, should be “against”, please correct.
Response: thanks for the deep revision, it has been changed as indicated.
3. Abbreviation for Coenzyme Q sometimes shows up as “CoQ” and sometimes as “COQ”. Please be consistent and implement corrections.
Response: Thanks for the deep review. We have check the text and discriminate into CoQ as the molecule and COQ(italics) as the genes or COQ for proteins.
4 Line 56 and line 59, shows up the same generic statement “many diseases”. It doesn’t read well, please correct.
Response: Thanks for the recommendation. We have changed the sentence for clarification. Now it is as follows: "Furthermore, many other diseases can show secondary CoQ deficiency although it is not associated with direct dysfunction in enzymes involved in CoQ biosynthesis [14]."
5. Line 65 “Polyphenols and resveratrol…” this statement is misleading as it suggests that resveratrol is not a polyphenol. Please re-write.
Response: Thanks for the recommendation. The sentence now starts as "The stilbene polyphenol resveratrol (RSV) and many other polyphenols..."
6. Lines 76-79, this sentence is too long and doesn’t read well, therefore the meaning is lost. Please re-write.
Response: Thanks for the recommendation. We have re-written the sencente accordingly with your comment and the comment of other reviewer as follows: "We have previously found that RSV affects CoQ levels and activity of CoQ-dependent oxidoreductases in old animals in an organ-dependent effect [28,30,33]. In the present study, we wanted to know if the RSV effects in obese animals also include the regulation of CoQ synthesis and the modulation of COQ genes and if this regulation is associated with the modulation of the mito/autophagy system."
7. Line 194, the abbreviation TAS is not explain. Please explain all non-obvious abbreviations fist time, when they show up in text. Moreover, this abbreviation refers to Figure 1E, “Total antioxidant activity”, shouldn’t it be “total antioxidant status”?
Response: Thanks again for the recommendation, you are right. We have introduced the meaning of the abbreviation as suggested. Now, in section 2.2, we have introduced the change: Total antioxidant status (TAS) and changed activity for status in the legend of figure.
8. Figure 2, the wording in the sentence “B-actin was used as a housekeeping gene” is unfortunate, since B-actin is housekeeping gene, and was used as a loading control in this study. Please re-word.
Response: Thanks for the recommendation. This sentence have changed to: "The housekeeping gene β-actin was used as loading control."
9. Starting from line 225 the new abbreviation “HDF” shows up. Do the authors mean high fat diet (HFD)? Again, abbreviations should be consistent. Please correct or explain new abbreviation.
Response: Sorry for the mistake, HDF is HFD. This has been corrected and also in the images.
10. Line 240, what does “se” mean? Please correct.
Response: Sorry for the mistake, it has been corrected.
11. Figure 6, please explain, what HDF stands for.
Response: As indicated in comment 9, HDF was a mistake and has been changed for HFD.
Reviewer 2 Report
The manuscript to provide a rationale for used RSV to mitigate damages associated to metabolic diseases and aging. In particularly, the authors state that the rsv modulates the levels of PINK1 and PARKIN mitophagy markers. The paper is a potentially interesting manuscript. The design of the manuscript is clear and the bibliographic citations adequately support their final conclusions.
Major revision:
!)To validate the role of resveratrol in mitophagy it would be useful to also determine other markers such as LC3I and LC3II. 2) The authors should add in addition to the analysis of western blots also immunofluorescence images of liver tissue to highlight the mitochondria
Author Response
We thank the reviewer for his/her comments that enrich our manuscript.
1)To validate the role of resveratrol in mitophagy it would be useful to also determine other markers such as LC3I and LC3II.
Response: Thanks for the recommendation. Unfortulately, we cannot perform the determination suggested since due to the lockdown for COVID-19 pandemy. However, we have introduced other markers that reinforce our the evidence that RSV is inducing mitophagy in liver from HFD-fed animals. We have introduced BNIP3L-NIX marker in mitochondrial extracts (figure 6) that is known as component of the mitophagy machinery and Beclin-1 and autophagy related 3 (ATG3) as markers of the autophagic machinery (figure 7).
2) The authors should add in addition to the analysis of western blots also immunofluorescence images of liver tissue to highlight the mitochondria.
Response: Unfortunately, we have not samples processed for immunohistochemistry to perform these assays. All the liver has immediately frozen after removal. However, we consider, data from WB determination are enough to demonstrate the importance of autophagy in RSV effect.
Reviewer 3 Report
The article by Catherine Meza-Torres provides a very interesting a promising research on the effect of resveratrol in CoQ10 biosynthesis in murine models on HFD. The methodology proposed is sound and the conclusions of the authors appropriate. Nonetheless the article could be improved. In this respect I propose some issues that should be addressed by the author in a revised versione and some minor aspects mainly related to the english language and some typos or imprecisions scattered in the text.
Major Issues
Section 2.1 no consideration on sample size and statistic, how did the authors address the number of animals in the study?
Paragraph 3.3 from line 223 Why among all the CoQ-synthome specifically COQ4 and COQ7 only were verified at protein level? Equal amount of mitochondrial proteins were loaded, which is reasonable, but in general there are any indications of variations in mitochondrial mass? Overall cellular content might have had a different outcome. The same consideration apply to section 3.6. This aspect should be discussed and markers of mitochondrial biogenesis/mitochondrial mass would significantly contribute to the paper.
Figure 4 and chapter 3.4 The author used electrochemical detection. Consideration on CoQ oxidative status would also be useful, in particular since the authors speculate on increase in Q10 due to the antioxidant form. Could this data be added?
Minor issues
Introduction
Line 48 not clear “Recently, a plasma membrane outer side linked oxidoreductase enzyme has been associated with the reduction of CoQ located at lipoproteins in blood plasma”
At line 55 I would specify "both variants and mutations" with different biological consequences
At line 57 modify "CoQ proteins" with “enzymes involved in CoQ biosynthesis”
At line 75 “Previously others and we found that…” should be rephrased
At line 78 it is reported the aim of the article mainly focussing on “the regulation of CoQ synthesis and the modulation of COQ genes" However the authors have also investigated aspects of mitochondrial dynamic and oxidative status that should also be mentioned in this final phrase.
Table one reports Anti‐CYTB5R Ab with a reference to an article, were those not commercially available Ab? In case they were not the correct reference should be used
At line 240 check spelling “ CoQ10 is associated with …”
Figure 5A it is not clear what TRI stands for, Probably should be HFD, same in TRI+RESV should be consistent with the rest of the manuscript HFD+RSV
Discussion line 295-297 The meaning is understandable but it should be rephrased because it sounds very colloquial and with some repetitions.
Author Response
Thanks for the recommendation. We hope, changes introduced in the text will improve it enough to be accepted for publication.
Major points:
1.- Section 2.1 no consideration on sample size and statistic, how did the authors address the number of animals in the study?
Response: we followed the Three Rs for animal research: Replacement, Reduction and Refinement and determined that four animals per group was enough to perform this determination. Previous studies about HDF changes indicated that in many of the parameters, four animals were enough to reach statistical differences. For this reasons and to use the low number of animals possible, we determined that 4 animals per group were enough.
2. Paragraph 3.3 from line 223 Why among all the CoQ-synthome specifically COQ4 and COQ7 only were verified at protein level? Equal amount of mitochondrial proteins were loaded, which is reasonable, but in general there are any indications of variations in mitochondrial mass? Overall cellular content might have had a different outcome. The same consideration apply to section 3.6. This aspect should be discussed and markers of mitochondrial biogenesis/mitochondrial mass would significantly contribute to the paper.
Response: We thank the reviewer for this interesting question. We used COQ4 and COQ7 proteins for two reasons. First, we have confirmed the specificity of these antibodies in KO cells for both proteins. In the market there are commercial antibodies that show no clear specificity. For this reason, we have used these antibodies in previous publications (Spinazzi et al., 2019). Secondly, in mRNA levels determinations, both proteins showed a different behaviour with reversal of the decrease induced but HFD in the case of COQ4 and no changes in the case of COQ7. For these reasons we used these proteins as markers.
Regarding mitochondrial mass markers, we show in figure 6 the determinations from crude mitochondrial fractions. in this case, the amount of mitochondrial mass markers is directly related to protein loading. However, to address the question of the reviewer we have performed the determination of VDAC1 in whole homogenate (figure 7). In this determination we found that HDF and HDF+RSV decreased the amount of VDAC1 indicating a reduction of mitochondrial mass.
Regarding the levels of mitochondrial biogenesis markers, we apologize but at this moment with the lockdown because COVID-19 we are unable to perform more determinations. The determinations introduced in Figure 7 and also in figure 6 (BNIP3L/NIX) were performed after asking for special permission. However, we think that the whole determinations indicate modifications in mito/autophagy and induction of this in the case of RSV.
3. Figure 4 and chapter 3.4 The author used electrochemical detection. Consideration on CoQ oxidative status would also be useful, in particular since the authors speculate on increase in Q10 due to the antioxidant form. Could this data be added?
Response: these data are unavailable. In our routine determination of CoQ we use the electrochemical detection because it is more sensible than UV detection. In our determination we oxidize all the CoQ present in the sample for determination, for this reason, we determined the whole amount of CoQ without discriminating between the redox forms.
Minor issues
1. Introduction. Line 48 not clear “Recently, a plasma membrane outer side linked oxidoreductase enzyme has been associated with the reduction of CoQ located at lipoproteins in blood plasma”.
Response: This sentence has been changed for: "Recently, a new oxidoreductase enzyme linked to the outer side of plasma membrane of hepatocytes has been discovered to reduce CoQ located in blood plasma lipoproteins [10]."
2. At line 55 I would specify "both variants and mutations" with different biological consequences
Response: changed as indicated, now in line 56.
3. At line 57 modify "CoQ proteins" with “enzymes involved in CoQ biosynthesis”.
Response: Changed as indicated. Although this sentence has been modified completely because a comment of other reviewer. Now it is as follows: Furthermore, many other diseases can show secondary CoQ deficiency although it is not associated with direct dysfunction in enzymes involved in CoQ biosynthesis [14].
4. At line 75 “Previously others and we found that…” should be rephrased
Response: changed for "RSV also changes mitochondrial physiology in mice under HFD conditions [20,21]."
5. At line 78 it is reported the aim of the article mainly focussing on “the regulation of CoQ synthesis and the modulation of COQ genes" However the authors have also investigated aspects of mitochondrial dynamic and oxidative status that should also be mentioned in this final phrase.
Response: This sentence has been changed for: "In the present study, we wanted to know if the RSV effects in obese animals also include the regulation of CoQ synthesis and the modulation of COQ genes and if this regulation is associated with the modulation of the mito/autophagy system"
6. Table one reports Anti‐CYTB5R Ab with a reference to an article, were those not commercially available Ab? In case they were not the correct reference should be used.
Response: This was a non-commercially available Ab. We have included a sentence in Materials and Methods section: "For anti-CYTB5R determinations we used a rabbit polyclonal antibody gently provided by J.M. Villalba group [35]. "
7. At line 240 check spelling “ CoQ10 is associated with …”
Response: This sentence has been changed (is instead of se). Sorry for the mistake.
8. Figure 5A it is not clear what TRI stands for, Probably should be HFD, same in TRI+RESV should be consistent with the rest of the manuscript HFD+RSV.
Response: Thanks for the comment. This was an internal manner to indicate HFD and now it has been changed.
9. Discussion line 295-297 The meaning is understandable but it should be rephrased because it sounds very colloquial and with some repetitions.
Response: This section has changed in order to clarify the message. It is as follows: "The clearest relationship of RSV with CoQ was found when RSV was considered a putative precursor for CoQ synthesis in yeast [49]. However, in animals, and particularly in mammals, this fact seems to be very unlikely and RSV cannot be considered a source of the phenol head of CoQ [32,49]. Furthermore, RSV and COQ have also shown negative interactions since RSV has been shown to interfere in the transit of CoQ-delivered electrons between mitochondrial complexes I and III [50]."
Round 2
Reviewer 2 Report
The manuscript can be accepted in this form